# Key Strategic Decisions and Their Influences on the Management and Success of the Bank of America Chicago Marathon and the Marathon Valencia Trinidad Alfonso

**Juan L. Paramio-Salcines** [1,*] **and Ramón Llopis-Goig** [2]

1   Department of Physical Education, Sports Science and Human Movement, Universidad Autonoma de Madrid, Campus de Cantoblanco, Francisco Tomás y Valiente, 3, 28049 Madrid, Spain

2   Department of Sociology and Social Anthropology, Universidad de Valencia, Avda. Blasco Ibáñez, 13, 46010 Valencia, Spain

*   Correspondence: juanluis.paramio@uam.es

**Abstract:** City marathons have evolved and grown exponentially in type and popularity, in their managerial complexity, and in terms of their financial impact on their host cities and the attraction of corporate sponsors. Most of the research on city marathons has focused on evaluating their broad economic, urban, tourist, social, sporting, and symbolic effects on host cities. However, less attention has been paid to analyzing key strategic decisions that could account for the evolution and growth of specific marathons and their influences on their management and success. This article, which addresses the cases of the Bank of America Chicago Marathon and the Marathon Valencia Trinidad Alfonso, examines those key strategic decisions that have been taken from their inaugural first editions to present and how effective they have been as regards the management and success of both races. Results show that the international success of both events –in terms of sporting participation, performance, and economic impact– is closely related to critical key decisions taken to improve the design and management of the event; the synergies between the political, business and sporting spheres that the organizational leadership of both races has favored their implementation and, as a consequence, the support received from sponsors. This factor has not only provided both races with financial stability, but it has also contributed to improving how both marathons are managed.

**Keywords:** city marathons; key strategic decisions; corporate sponsors; the Bank of America Chicago Marathon; Marathon Valencia Trinidad Alfonso

## 1. Introduction

In the late twentieth century and contemporary period, running in city marathons worldwide has become a dynamic and specialized market of the sport and leisure industry. As part of what we describe as the globalization of city marathons and within a hyper-competitive market at national and international level (Llopis-Goig and Paramio-Salcines), these races have continued to evolve and grow in number and popularity as mass events in all developed and developing countries alike (Scheerder et al. 2015 referred to it as the second wave of running), for their potential to attract corporate funding and, not least, for their broad economic impact on their host cities. As part of a consolidated running boom globally, in their annual report, World Athletics (2021), the international governing body for athletics, estimated that there are 552 certified marathon races around the world, including eight in Spain (e.g., Valencia), eleven in the UK (e.g., London) and thirty-three in the U.S. (e.g., New York, Boston and Chicago). Focusing on the U.S. market, academics and industry reports estimated the significant growth of city marathons from fewer than 40 marathons in 1969, to nearly 200 marathons in 1977 (Cooper 1992), 850 marathons in 2012, to a consolidated 1100 marathons in 2017, attracting over half a million finishers (Running USA, in Miller and Washington 2017; Statista 2018).

As a result, planning, executing and managing contemporary city marathons has become more complex within, as mentioned, a hyper-competitive market. Western and new global actors such as Brazil, Russia, India, China and South Africa (known by the geopolitical acronym of BRICS countries) and Middle East cities have developed the "mega sport event strategy" over the years (see, for instance, Burbank et al. 2001; Llopis-Goig 2012; Müller 2015; Roche 2017; Zimbalist 2015), so that hosting a city marathon, more than just an athletic event, has become a highly relevant urban, economic, tourist, sporting and symbolic strategy for cities seeking the world's attention in this contemporary period. In a quest to understand the economic motivation of cities for hosting marathons, Running USA, a non-profit organization committed to the growth and success of the running industry, stated, "cities are embracing marathons for the economic upswing. One of the benefits of a marathon of any size is that it brings people to your city, it showcases your city, it brings people back" (in Miller and Washington 2017, p. 277). Andrew Suozzo (2006, p. 112) went further by succinctly stating that "in many ways, a city without a major marathon is a city that has not arrived". In this process, many city marathons have also bid for the attraction of corporate sponsors, which has contributed, among other things, to the financial stability of the race (Cooper 1992), to the attraction of top male and female runners (Frick et al. 2019), and in some cases to apport their brand to the name of the race (Paule-Koba 2020b).

Evaluation of the broad economic, urban, tourist, social and sporting effects of city marathons on their host cities has attracted more attention since the beginning of the new millennium Not surprisingly, this international phenomenon has received significant attention from academic and industry researchers of very different disciplines such as sports and leisure economics (Cobb and Olberding 2007; Coleman 2004; Coleman and Ramchandani 2010; DeSchriver et al. 2022; Foster et al. 2021; Frick et al. 2019; Gratton et al. 2006), sport and event management (Huang et al. 2015; Llopis-Goig and Paramio-Salcines 2021; Paule-Koba 2020a; Shipway and Jones 2008; Thomson et al. 2019; Wicker et al. 2012), economics and business administration (García-Vallejo et al. 2020), sport history (Cooper 1992, 1998), sociology (Capsi and Llopis-Goig 2021; Scheerder et al. 2015), tourism (Gibson et al. 2012; Melo et al. 2021), health science (Reusser et al. 2021) and even cultural studies (Suozzo 2002, 2006). In this process, city marathons have been consolidated as major tourist destinations by attracting large numbers of elite and amateur runners from outside the host city and abroad (e.g., Gibson et al. 2012; Hallmann and Wicker 2012; Heere et al. 2019; Huang et al. 2015; Park et al. 2019; Wicker et al. 2012).

Although Foster et al. (2021, p. 428) remark that "creativity and innovation play a more important role" to explain the evolution of a mega-event such as the Superbowl, less attention has been paid to analyzing the key strategic and innovative decisions that could explain the evolution and growth of specific marathons (see Cooper 1992 who analyzed the decisions taken by Fred Lebow, race director at the New York City Marathon) and by extension, the contribution of sponsors to the financial stability and their influence on the management and success of those races, an issue highlighted by Suozzo (2006, p. xv) who said that "despite the current marathon craze, the race (the Chicago Marathon) itself has been the object of little serious study". Suozzo has been one of the few academics to analyze the broad effects of the Chicago Marathon, not only on the urban renaissance of the city, but also to identify the strategic decisions taken over time that explain the growth and the contribution of sponsors to the financial stability of this world marathon major as defined by the Abbott World Marathon Majors (AbbottWMM) organization (Abbott World Marathon Major 2022).

Adopting a broad interdisciplinary perspective, in this paper we analyze the cases of the Chicago Marathon (the Bank of America Chicago Marathon as it is currently branded), one of the six annual World Marathon Majors by AbbottWMM organization (including the Tokyo Marathon, Boston Marathon, TCS London Marathon, BMW Berlin Marathon and TCS New York Marathon), a certified race by World Athletics and also recognized as a Gold Label race by the International Association of Athletics Federations (IAAF) (Bank of America 2022), and the Valencia Marathon (now known as the Valencia Trinidad

Alfonso EDF marathon), one of the world's most renowned marathons, a certified race by World Athletics, recognized as a Platinum Label race by the IAAF since 2020 and valued as a benchmark of the second wave of running in Europe (Capsi and Llopis-Goig 2021). Both marathons are hosted in cities such as Chicago, the U.S.'s third largest city with a population of 2.74 million (United States Census 2021) and Valencia, Spain's third largest city with over 800,000 inhabitants and one of the leading destinations for national and international visitors. Both city marathons are not just valued for their short-term potential to generate broad direct and indirect sporting and non-sporting benefits (as well as, in some cases, negative effects) to their host cities, but also are seeking other intangible effects by reinventing themselves, as a major point of differentiation, as in the case of Valencia, as the "city of running" (Capsi and Llopis-Goig 2021; Llopis-Goig and Vilanova 2015) and Chicago as an "international city destination" (Suozzo 2006, p. 21).

This paper presents the main findings of both case studies based on content analysis of an extensive array of academic and professional publications, annual reports from both city councils, internal documents produced by both city marathons' organizers from their inaugural editions to nowadays (the Chicago Marathon, then the Mayor Daley Marathon), started officially, for the second time on 25 September 1977, while Valencia's first edition was on 29 March 1981), official race websites, annual reports undertaken by economic groups in those cities and personal communications with staff from the AbbottWMM and the Bank of America Chicago Marathon. Specifically, this paper analyzes the similarities and differences of the key strategic decisions taken by senior managers and race directors of both races over the time mentioned, with an emphasis on interdependent factors that influence management such as the type of organizational structure, the best sporting performance, ratio of elite and amateur runners, their economic, sporting, social and tourist impacts, and other indicators such as number, type, length and economic contributions from sponsors, estimated spectators, number of participants from all over the world, prize money to winners, and budget of the management of the race. The next section extends previous work (Capsi and Llopis-Goig 2021; Llopis-Goig 2012; Llopis-Goig and Paramio-Salcines 2021; Llopis-Goig and Paramio-Salcines) and examines related literature on the relationships between city marathons and their broad impacts on host cities. Section three explains the methodology followed and the criteria selected to choose both city mara-thons, which is followed by a section that provides a detailed discussion of the main findings and the managerial implications of this comparative international study for sport managers, city officials, and race organizers.

## 2. City Marathons Worldwide and Their Broad Effects on Host Cities

Research on city marathons is a relatively new phenomenon, but in parallel to a highly competitive market between cities at national and international level, it has been in constant expansion since 2000. As noted, much of the existing academic literature on this area has traditionally been devoted to analyzing the economic, urban, tourist, social and sporting effects on host cities; a trend that will continue in the coming decades. Not surprisingly, the academic literature has mainly evaluated the races as purely economic phenomena, with a particular focus on estimating the economic impact on their host cities. Although estimating the economic impact should be a relatively simple process, there is certain controversy about the type of methodology to be followed, either by ex-ante studies, concurrent or ex-post studies (see, for example, Agha and Rascher 2016; Barajas et al. 2012; Cobb and Olberding 2007; DeSchriver et al. 2022; Matheson 2012; Papanikos 2015; Zimbalist 2015). In the same way, there is no consensus regarding the impact of local people in sport events: some economists (e.g., Matheson 2012) suggest that any economic impact study should not include the expenditure of local people, while other economists (e.g., Cobb and Olberding 2007) suggest that those studies should include the expenditure of local runners in their local marathons as part of what they call the "import substitution". Other areas of interest have included the tourist impact of large numbers of recreational runners coming from outside the host city (Gibson et al. 2012; Heere et al. 2019; Wicker et al. 2012). Only in

recent years have studies integrated the key strategic decisions implemented by senior management of races either alone or in partnerships with urban leaders and corporate sponsors that could explain their evolution to become a world major marathon over time (Chicago Marathon) or world-renowned marathon (as the Valencia Marathon represents), an interrelation that is central within this paper.

Despite this progress, there are some limitations to what is currently known about marathons. There are, at least, four overarching topics: (1) there is no clear consensus among academics about precisely defining what type of event a city marathon represents; (2) how to correctly estimate the economic impact study of marathons; (3) what factors and key decisions from practitioners, rights holder organizations and organizers are critical to classify one marathon as major; and (4) the contributions of corporate sponsors to the financial stability and success of those events.

The AbbottWMM organization, created in 2006, values six, including the Chicago Marathon, of the world's major city marathons based on a variety of criteria, including the sporting performance, level of participation, economic impact, and other indicators (estimated spectators, sponsors, . . . ) (Abbott World Marathon Major 2022). Based on three of the best marathons worldwide, the TCS New York City, Boston and the Bank of America Chicago, Paule-Koba (2020a, p. 118) describes them as recurring events that happen on a regular (annual) time frame, understood as "the "easiest" type of events to execute because it occurs consistently". Other academics (Foster et al. 2021; Müller 2015) such as UK Sport, the government agency responsible for investing in Olympic and Paralympic sport, consider those races as "participant events" based on the high participant number of runners, or Papanikos (2015) described marathons as "sport tourism events". Indeed, Foster et al. used the high ratio of participation of runners to classify specifically three of the world major marathons (New York, Boston and London) as "global, high profile participation events". Apart from this, Martin and Hall (2020) focused on the level of sporting performance of participants to describe a city marathon as a world marathon major (see also (Frick et al. 2019) who argue that higher prize money paid to athletes attracts better runners). Focusing on the sporting performance, those authors' views coincide with the ranking of the six most significant world majors as proposed by the AbbottWMM organization. From a sociological perspective, Roche (2017) differentiates between (first order) mega-events, special events, hallmark and community types. Considering the complexity and variability of types of events in the second phase of modernization, Roche suggests that we should expand this typology by including second-order mega-events (see Müller 2015, in this debate over criteria to define what is a mega-event and an event).

However, we still know relatively little about what critical factors explain how city marathons can evolve to become world major marathons (the exceptions being studies by (Cooper 1992) on the New York City Marathon; (Suozzo 2002, 2006) on the Chicago Marathon or by (Capsi and Llopis-Goig 2021; Llopis-Goig and Paramio-Salcines 2021; Llopis-Goig and Paramio-Salcines), on the Valencia Marathon). Most of the research on this issue has been directed towards an understanding of the importance of developing strategic capabilities, as Collis (2021) remarks, in a holistic perspective rather than only focusing on particular issues. Johnson et al. (2017) value strategic capabilities to evaluate the critical factors of success in any business, while few scholars have focused on these issues on the management of events (Foster et al. 2021; Schwarz et al. 2017). Indeed, Foster et al. (2021); García-Vallejo et al. (2020); Newland (2020); Yoshida et al. (2013) and others argue that event concept innovations, co-creation and associated management issues have not yet been extensively studied. It is also relevant to examine the role of urban leaders, executive race directors (see (Cooper 1992) who examined the role of Fred Lebow in the New York City Marathon) or corporate sponsors on the success of races, with a few exceptions such as the studies undertaken by Suozzo (2002, 2006) on the Marathon of Chicago, issue that we analyzed later.

Most of the research on the managerial aspects, impacts and legacies of marathons on their host cities are based, not surprisingly, with the U.S. at the forefront (Cobb and

Olberding 2007; Cooper 1992, 1998; Fickenscher 2006, 2011; Martin and Hall 2020; Miller and Washington 2017; Suozzo 2002, 2006), followed by Britain (Coleman 2004; Coleman and Ramchandani, 2010; Gratton et al. 2006; Shipway and Jones 2008), Germany (Reusser et al. 2021; Wicker et al. 2012), China (Huang et al. 2015; Qiu et al. 2020), Greece (Papanikos 2015) or Spain (Alemany-Hormaeche et al. 2019; García-Vallejo et al. 2020; Parra-Camacho et al. 2021; Pedrosa-Carrera 2016; Pérez-González et al. 2021; Llopis-Goig and Paramio-Salcines 2021; Llopis-Goig and Paramio-Salcines).

Over the years, more financial data on some of the leading world city marathons have been made public. Indeed, from a comparative economic impact perspective on urban marathons, it has been reported that the direct economic impact of The SalleBank Chicago Marathon in Chicago was estimated to be USD 90 million in 2001 (Suozzo 2002) (a figure that in the 2013 Chicago Marathon was increased to more than USD 253 million in business activity as an independent study conducted by University of Illinois at Urbana-Champaign's Regional Economics Applications Laboratory estimated (Bank of America 2022)). This study found that the race directly contributed an estimated USD 88 million distributed among the main sectors of the tourism industry, plus another USD 131 million in indirect activity, accounting for more than USD 219.7 million worth of total business activity and an equivalent of 1520 full-time jobs and USD 74.58 million worth of wages and salary income. In a follow-up study, the University of Illinois at Urbana-Champaign's Regional Economics Applications Laboratory estimated USD 282 million worth of total business activity (Bank of America 2022).

One of the major competitors of the Chicago Marathon and global leader in marathons, the ING New York Marathon, brought USD 340 million to the city, drawing two million spectators and more than 45,000 runners in 2011 (Fickenscher 2011; see also New York Road Runners 2020). More recently, Germano and Cervantes (2015) noted that the TCS New York City Marathon (as it is now known) brought a bigger economic impact, estimated at USD 415 million to New York three years later in 2014. Beyond the U.S., Gratton et al. (2006) estimated that the London Marathon contributed GBP 1.2 million from participant spending in London. In a follow-up study on the Virgin London Marathon (now known as TCS London Marathon), Coleman and Ramchandani (2010) estimated that this leading marathon brought GBP 27.1 million into the London economy (GBP 13.1 million from spectators, GBP 9.5 million from runners, GBP 6.9 million from bed-nights in London hotels and guest houses, GBP 6.3 million from the catering industry and GBP 1.6 million from organizers' net spend).

In contrast to most studies that used ex-ante methodologies (described by Zimbalist (2015) as promotional studies), the Instituto Valenciano de Investigaciones Económicas (IVIE), following an ex-post methodology, estimated that the operating costs of the Valencia Marathon amounted to EUR 5.4 million, while the revenue generated by the money spent in Valencia by runners and their companions amounted to EUR 22.8 million in 2019 (IVIE 2020; see also Instituto Valenciano de Investigaciones Económicas (IVIE) 2017, 2018, 2019). It is interesting to draw attention to these large differences in the economic impact studies. Sport economists such as Agha and Rascher (2016); Agha and Taks (2019); Barajas et al. (2012); DeSchriver et al. (2022); Matheson (2012) and Zimbalist (2015) argue that these differences are explained by the methodology used to measure the economic impact of hosting urban marathons.

Wicker et al. (2012) studied the key driving factors that influence the consumer expenditure of runners and their intention to revisit three marathons in Germany (Cologne, Bonn, and Hanover) (see also Hallmann and Wicker 2012; Park et al. 2019). Similarly, tourism scholars, Huang et al. (2015), used the 2012 Shanghai International Marathon to add a new perspective as represents the image congruence between marathons and host cities. Martin and Hall (2020) specifically explored the impact of the New York City Marathon on the city's hotel demand after Hurricane Sandy in 2012 which contributed to the cancellation of the race. In Spain, as noted, the city marathons' literature has recently expanded, by incorporating some studies such as García-Vallejo et al. (2020),

which examined five marathons in Spain (Barcelona, Madrid, Malaga, Sevilla and Valencia) from the management perspective after identifying the main organizational areas and how to maximize the planning and management of those races. Pérez-González et al. (2021) analyzed different aspects from the economic impact and the level of satisfaction of runners to the return from sponsors of the 2019 Burgos Marathon. Parra-Camacho et al. (2021) focused specifically on the social impact of the Valencia Marathon from the residents' perceptions, while Pedrosa-Carrera (2016) analyzed the management factors at the 2015 Zurich Seville Marathon. As with the study of Wicker et al. (2012) in German marathons, Alemany-Hormaeche et al. (2019) examined the segmentation of runners and their level of satisfaction in the 2016 Palma de Mallorca Marathon.

## 3. Method

By comparing both cities' marathons as case studies, we sought to identify those aspects of management, innovation, political and organizational leadership, and development that have been decisive in the growth and success of both events over time and what are, if there are any, the competitive advantages of both marathons.

In the case of the Chicago Marathon (originally known as the Mayor Daley Marathon in honor of Mayor Richard J. Daley who passed away nearly one year before the inaugural race in 1977), this evolution has taken less time than the Valencia Marathon as the race became a premier international event before the turning of the new millennium (Suozzo 2002, 2006). As an example of high level of participant event, the Chicago Marathon has been valued as the second top marathon in the U.S. for number of finishers (45,932) in 2019 (Bank of America 2022), the third largest race by number of finishers at global level after the New York and Paris Marathons (Statista 2021), the fourth marathon for its economic impact (USD 80 million) and the second for prize money paid to athletes (Frick et al. 2019). As part of its growth in participant and sporting performance, the Chicago Marathon (now known as the Bank of America Chicago Marathon) became part of the Abbott World Marathon Majors in 2006. Unlike the Chicago Marathon, the Valencia Marathon has taken forty years to evolve from a community city marathon race to be one of the world-renowned marathons. This evolution has contributed to situating the current Valencia Marathon (now known as the Marathon Valencia Trinidad Alfonso EDP) at the top of ranking of participants in marathons in Spain, the fourth marathon in Europe (after Berlin, London and Paris) and the seventh at global level. In addition, as a competitive advantage, the Valencia Marathon has achieved the sixth fastest time in the history of the men's marathon, beating the Chicago Marathon. These data and the background make the two marathons comparable. Moreover, in demographic terms, the two cities have certain equivalencies. Both rank third in size in their respective country rankings, although the city of Valencia barely reaches one million inhabitants, while Chicago has a registered population of 2.74 million.

To conduct the study, the paper provides some historical background and perspective of both marathons and to their main evolutionary milestones and key decisions in terms of management and the contribution of corporate sponsors to the financial stability and growth of both races. As noted earlier, the analysis was carried out through secondary sources, especially economic impact studies, as well as scientific literature, documentary sources, management reports and official race websites providing information on both events and personal communication with staff from the AbbottWMM and the Bank of America Chicago Marathon organizations. In the case of the Valencia Marathon, annual economic impact studies of the Instituto Valenciano de Investigaciones Económicas (Instituto Valenciano de Investigaciones Económicas (IVIE) 2017, 2018, 2019, 2020) were particularly useful, while for the Chicago Marathon, the work of Suozzo (2002, 2006) was a fundamental source of information.

The aspects analyzed refer to the evolution in terms of participation, sporting performance, economic and tourist impact and, especially, to six key interrelated aspects of the management of both events: the design of the course, organizational leadership, the contribution of sponsorship to the financial stability and to the growth of the race, the prize

bonus paid to elite athletes and their effects on the sporting performance, political and organizational leadership and the fit with the city's promotional policies and the connection of the event with the local sporting environment and culture. Before addressing these aspects, we followed the recommendation of Fox et al. (2014) (see also Burbank et al. 2001) when they note that in conducting research on events, it is critical to examine the political and socio-economic context of the event itself to shed light on their policy implications.

*The Political and Economic Context of Chicago and Valencia*

The Chicago Marathon and the Valencia Marathon have been developed in a context marked by a series of certain political and socio-economic parallels, which we report on in the following lines. Both have been developed in a period of severe economic and social difficulties, which have been managed by municipal public policies.

The Chicago Marathon has experienced extraordinary growth in parallel to the recovery that the city of Chicago had to face in a context in which its future was in serious doubt. This growth has benefited from the support received by different mayors in the city since the 1970s, from the financial contribution of sponsors since 1979 onwards as well as the event's fit with the kind of urban actions and transformations undertaken by the city. Shortly before the first edition of the Chicago Marathon, the city was paralyzed by the deaths of two mayors within a short period of time, including Mayor Richard J. Daley in December 1976 and later, the first and highly charismatic African American mayor, Harold Washington (1983–1987) (Suozzo 2002, 2006). Alongside the problems of industrial decline, Chicago faced a strong racial polarization that questioned its ability to retain its own population and, moreover, to attract potential tourists. In the post-1976 period, under the new leadership of Mayor Jane Byrne (1973–1983), and followed by Richard M. Daley (1989–2011), the Chicago City Council realized that the city had a negative image, strongly associated with crime and squalor, and therefore considered it necessary to reposition it as "an international city" destination (Suozzo 2006). In this complex process of image building, the Chicago City Council led an ambitious urban regeneration program that relied on urban design by optimizing sidewalks, creating green areas and urban parks, and redesigning public spaces, buildings and access to the local metro. As with other cities around the world (Roche 2017), the Chicago City Council not only provided architectural icons for the city's major neighborhoods and communities (one of the latest architectural developments was the Millennium Park, designed by the Canadian American mega architect Frank Gehry), but also devoted public resources to ensuring that race, ethnicity and sexual orientation were seen as valued and distinctive identities (Suozzo 2002).

Another key urban project was the urban regeneration of one of the oldest parks in the city, Grant Park, the greatest urban space and a "green belt" connected to the Chicago city center and linking various local cultural infrastructures and leisure centers. Over the years this central park has hosted numerous cultural and sporting events and urban festivals. After Carey Pinkowski became executive race director of the Chicago Marathon in 1990, one of his critical decisions was to change the starting and finishing line of the Chicago Marathon to the central and spacious Grant Park (Suozzo 2006). Sport has played an important role in the transformation and reinvention of Chicago's image, and this has been influenced by the success of Michael Jordan's NBA Chicago Bulls, followed by the "mega sporting event strategy", despite having a failed bid for the 2016 Summer Olympic Games, and the success of the Chicago Marathon. In the case of Valencia, after more than a decade (1980s) of local government under the control of the social democratic party, the symptoms of economic and industrial decline began to be evident in the city. As an alternative strategy, local actors considered the tourism sector, for which the city was not prepared despite its coastal location. Influenced by the success of Barcelona with the 1992 Olympic Games, it was at this point that Valencia, as previously mentioned in the case of Chicago, began to build high profile urban projects, located in the final section of the old riverbed of the Turia River, that would help to redefine the image of the city as an attractive tourist destination. Although some of the urban projects had begun previously, it was at the beginning of the

1990s with the leadership of Rita Barberá, of the conservative party, when two essential infrastructure projects for the city were promoted (Llopis-Goig and García Alcober 2012).

The reorientation of the political culture of Valencia has also been reflected in local sport policy. After following the mega sport strategy, as in Chicago and other cities worldwide, Valencia has hosted a wide portfolio of national and international sporting events throughout the calendar year, ranging from one-off events (such as the 32nd and 33rd edition of the America's Cup, a Formula One Grand Prix and the Open Tennis 500 tournament) to "recurrent events" such as the Valencia Marathon. However, this mega sport events strategy followed by the Valencia City Council and Valencia regional governments came to a dramatic halt at the beginning of the second decade of the 21st century when the effects of the Great Recession on Spanish society were undeniable. After that, local politicians started to value the potential of the Valencia Marathon in their overarching economic and tourist development strategy. The Valencia City Council proposed to the race organizers (Sociedad Deportiva Correcaminos) that they join forces with the aim of making this event more renowned internationally. After the approval of the 2011 Valencia Sports Strategic Plan, both stakeholders implemented several key strategies that led gradually to an extraordinary growth in status and international reputation as one of the world-renowned marathons. City Council support for this race was not altered by the arrival of a coalition of left-wing and regionalist parties in the City Council in 2015, which ended 24 years of conservative hegemony, even though these parties had previously been highly critical of the mega-sport event strategy. Since then, the Valencia Marathon has continued to improve in terms of the management of the race, the sporting performance and financial stability and the quality of services for runners (Llopis-Goig and Paramio-Salcines) and, not least, it has contributed to boost running at local level as the case of the Valencia Marathon illustrates (Capsi and Llopis-Goig 2021; Llopis-Goig and Vilanova 2015).

## 4. Results

As stated previously, the planning, execution and operation of a city marathon is a crucial part of a long-term strategic management process, and financial viability becomes a fundamental principle in the operation of the events. Taking into consideration that there is not a linear process to the implementation of decisions that lead to success (Collis 2021), Table 1 summarizes some of the key decisions adopted by urban leaders as well as organizers and race directors of both the Chicago Marathon and the Valencia Marathon over time. These decisions have also affected operational decisions linked to the evolution in size, status, location of the start and finish lines, timelines, financial contributions of sponsors, budget of the race, prizes bonuses paid to elite athletes and legacies of both marathons in their progression toward their status today as, arguably, two of the world's major marathons. As events move from amateur to mega-events, race directors and senior managers must adjust, as Collis (2021) suggests, to the changing environment and analyze the event holistically exploring the interrelations between different areas as part of the "complete landscape strategy" (see also García-Vallejo et al.'s 2020 study on the management process of the five main marathons in Spain). Foster et al. (2021) also consider that creativity and innovation play a more important role in understanding the evolution and expansion of both city marathons over the years. Not least, another key factor for successful event management is effective political and organizational leadership, which must facilitate all stages in the event lifecycle (Schwarz et al. 2017). In the case of the Chicago Marathon, Suozzo (2006) identified local actors such as different city mayors' support over the years, local enthusiastic organizers who were influential in the first edition of the Marathon, the financial contributions of individuals (Lee Flaherty and his company Flair Communications in the first two editions) and, mainly, the financial contributions from big corporate sponsors (Table 1), and, not least, the decision to incorporate different race directors (from Bob Bright in 1982 until 1986, when he was forced to step down for financial mismanagement, to Carey Pinkowski in 1990 to the Chicago Marathon organization who have been influential in the further economic, participation and financial stability growth

of the race. Both of these race directors have been vital in the attraction of elite male and female runners who have contributed to increase the race's reputation by setting world records (Bank of America 2022). As with the case of Pinkowski in the Chicago Marathon, the figure of Francisco Borao, the executive race director in the Valencia Marathon as well as CEO of the Association of International Marathons and Distance Races (AMIS), and the incorporation of the Foundation Trinidad Alfonso as the main sponsor of the race in 2013, have been regarded as the main turning points in the further international development of the Valencia Marathon.

**Table 1.** Selected key decisions that explain the growth and evolution of the Chicago and Valencia Marathons.

| Milestones in the History of the Chicago Marathon | Milestones in the History of the Valencia Marathon |
| --- | --- |
| 1977. The first edition of the Chicago Marathon (modern era) (known as Mayor Daley Marathon) on 25 September with 4200 runners and 2128 finishers<br>1979–1986. Beatrice Foods became the main corporate sponsor contributing to the financial stability of the race and increased the elite athletes prize money over the years, from USD 77,000 (1982) to USD 250,000 (50,000 more than New York race) in 1984 and to USD 285,000 in 1986<br>1983. Achieved its status as one of the America's most popular marathons and was renamed as America Marathon<br>1987. The Chicago Marathon had to be reduced to a half marathon, after the withdrawal of Beatrice Foods as the main sponsor<br>1988. The marathon was moved to the spring with Old Style (Heileman Brewing Company) being the main sponsor (1988–1990)<br>1990. Carey Pinkowski became the race director; identified as critical in the evolution and expansion of the Chicago Marathon<br>1991–1993. The race has no sponsors<br>1993. Major Events bought the Chicago Marathon and after that, its owner, Chris Devine, persuaded La Salle Banks to become the main sponsor<br>1994. The race became known as La SalleBank Chicago Marathon attracting other product sponsors such as New Balance, Sparkling Spring, Gatorade, Nextel and Volkswagen<br>1996. LaSalle Bank bought the Chicago Marathon from Mayor Events, Inc.<br>2001. The race changed its timeline to Columbus Day<br>2006. The race became one of the five selected World Major marathons by World Marathon Majors (WMM) (known as the Abbott World Marathon Majors)<br>2007. The race reached its cap of 45,000 entrants<br>2007. Bank of America bought LaSalle Bank's parent company ABN AMRO and the race's title sponsorship became as branded today "The Bank of America Chicago Marathon"<br>2008. Steve Jones broke the world men's record in 2:08:05<br>2013. The world men's record was set in 2:03:45<br>2019. The world women's record was set in 2:14:04, which is still the women's course record with 45,932 runners<br>2020. The race was cancelled due to the COVID-19 global pandemic | 1979. Establishment of the Club Sociedad Deportiva Correcaminos (SDC), organizers and right holders of the Valencia Marathon<br>1981. The first edition of the Valencia Marathon (Marathon Popular de Valencia), organized by the SDC, brought entries of 800 runners<br>1986. The Valencia Marathon incorporated into the Association of International Marathons and Distance Races (AIMS).<br>1988. Valencia hosted the Spanish Marathon Championship (also in 1993 and 2000 editions)<br>1989. The Valencia Marathon adopted a new, mostly urban, and flat course.<br>1995. The system of compensating finishers was established (also applied in the 1996 edition and abandoned in 1997)<br>2005. Launching of the popular Racing Circuit of Valencia.<br>2010. Francisco Borao (President of the Sociedad Deportiva Correcaminos since 2005) became the chairman of AIMS and re-elected in 2014 and 2018.<br>2011. Approval of *The Valencia Strategic Plan for Sport*. The race date was moved to the fall and the 10K race was added to the event. The start and finish lines were moved to the Ciudad de las Artes y las Ciencias. Divina Pastora became the main sponsor of the race.<br>2013. After becoming a sponsor in 2012 of the Valencia half marathon, the Fundación Trinidad Alfonso (Foundation Trinidad Alfonso) became the main sponsor of the event and contributed to building the race brand (MVTA EDP).<br>2014. Valencia City Council and the Fundación Trinidad Alfonso created the Valencia City of running organization, which brought together different races (around 50 with around 240,000 participants) held in the city (four of them holding "World Athletics certification").<br>2015. The launching of the 5K Circuito Jardin del Turia in the old riverbed of the Turia River<br>2016. Recognition of the Valencia Marathon as Gold Label Road Race (World Athletics).<br>2019. The Valencia 10k disappeared after 10 editions (from 2011 to 2019)<br>2020. Recognition of the Valencia Marathon as Platinum Label Road Race (World Athletics). An unexpected issue as the health restrictions caused by the COVID-19 global pandemic required a critical decision by the organizers. The 40th edition of the 2020 MVTA EDP was limited to elite-level runners only<br>2022. For this edition, the Valencia Marathon has 30,000 registered runners |

Source: Updated own elaboration based on different official sources (Bank of America 2022; see also Llopis-Goig and Paramio-Salcines; Suozzo 2002, 2006).

### 4.1. Who Have Been the Valencia and Chicago Marathon Rights Holders and Organizers over the Years?

The Marathon Popular de Valencia emerged in 1979 when a group of local runners set up the club Sociedad Deportiva Correcaminos (SDC), the oldest club in the city, with the mission to promote athletic events and to boost local athletics. Two years later, the first Marathon Popular of Valencia took place officially in 1981. The race was set up by a local couple, Miguel Pellicer and Angelita Carrasco, enthusiastic runners who were key in setting up the club (Lastra 1984) (Table 1). From a comparative international marathon runners' perspective, it is relevant to note that the first Valencia Marathon attracted only 800 runners, which was very low compared, for example, to the first London Marathon, celebrated the same day and year, which attracted 7055 runners (Llopis-Goig and Paramio-Salcines). Looking in perspective, although the Chicago Marathon was held from 1905 to the 1920s, the first edition of the present times took place on 25 September 1977, attracting 4200 runners (Suozzo 2002, 2006; Bank of America 2022), making this number of runners the largest marathon in the world at that time.

Coincidentally with the growth in running races in Spain (Capsi and Llopis-Goig 2021; Llopis-Goig and Vilanova 2015), the inaugural Marathon Popular de Valencia was officially held three years later than three of the main Spanish marathons held in 1978: the Marathon Popular de Madrid (April), Barcelona (March) and San Sebastian (October). Unlike the Chicago Marathon, the Valencia Marathon has been owned and organized—and continues to be, along with other local athletics races—by SDC since 1981. In the following year (1982), one of the first strategic decisions taken was to shift the official time schedule of the marathon from March to February, a decision which had been operative for over three decades until the organizers decided in 2011 to move the timeline of the race to November and, lately, to the first Sunday of December (editions of 2020, 2021 and for the forthcoming 2022). This decision has proved to be crucial in explaining the current success of the Valencia Marathon; not only did it allow the Valencia Marathon to distance itself from the calendar dates of the other top marathons in Spain, for example, Madrid, Barcelona and Seville, but also, the new date was more appealing for prospective elite and amateur runners due to the better race-day temperatures of the city and the timing in the preparation stage.

Equally important has been the range of unique elements that have come to signify the Chicago Marathon and the Valencia Marathon as winning business products. As part of this process, unlike the Chicago Marathon organizational structure, SDC—who are both the MVTA rights holder and organizer—have, year by year, taken some strategic and innovative decisions—from the time schedule to the location of the start and finish line of the race at the City of Arts and Sciences complex, built by the Spanish mega start architect Santiago Calatrava, in the center of the city (similar to the location of the starting and finishing line at the Chicago Marathon at Grant Park). The financial contribution of corporate sponsors has developed over the last decades (in the case of the Chicago Marathon) and over the last decade (in the case of the Valencia Marathon) (Table 1), as has the concept of the marathon as an entertainment event—which could explain the success of both marathons.

### 4.2. Corporate Sponsors and Their Financial Support to Both Marathons

Corporate sponsors are becoming more influential nowadays as they provide a relevant part of the funding of contemporary marathons. Drawing on the case of the Chicago Marathon's experience with sponsors over time and their effects on the financial stability of the race, Suozzo (2006, p. 89) drew attention to the important contribution of corporate sponsors and their effect on the financial stability of the race.

"All marathons are heavily reliant on corporate backing. Because spectators pay no fees, television revenues can be relatively modest, and running themselves finances only a small part of the actual cost of participation, race directors and owners must find other sources of income".

Similarly, and focusing on the cases of the five major marathons in Spain, including the Valencia Marathon, García-Vallejo et al. (2020) noted that "sponsors' contribution is crucial at the economic level: If there is no sponsorship, you can't bring good athletes, you don't break records. The reverse is also true: if the course is no good, sponsors don't come" (p. 9).

As the recent Bank of America's report shows, the history of sponsors in the Chicago Marathon has not followed a linear and easy process (Bank of America 2022). In fact, the original founders of the Chicago Marathon had no sponsor and, therefore, a local businessman, Lee Flaherty, had to cover the costs of the race itself. Individuals and corporate sponsors have been regarded, as Suozzo (2006) acknowledged, vital to ensuring financial stability to the race, and to changing its status and reputation by attracting the best elite male and female runners after increasing the prizes paid to top athletes, which have contributed to increase the sporting performance. After an initial difficult economic situation of the first years of the Chicago Marathon (originally known as Mayor Daley Marathon) and tensions between sponsors and runners regarding the running time and location, the situation changed after the incorporation of Beatrice Foods, a major American food processing company. According to the Bank of America's report (2022), the arrival of this company as the main sponsor became crucial to the financial stability and growth of the Chicago Marathon over the period (1979–1986) to become one of the best marathon races in the United States. One of the main and critical strategies, thanks to the race director of that time, Bright, was to increase the elite athletes' prize money, which contributed to the evolution of the race from a mass participation race to a race expecting to increase the sporting performance by attracting the best elite male and female runners (Bank of America 2022). Aligned with the analysis of Frick et al. (2019) and García-Vallejo et al. (2020), Table 1 shows that the decisions related to paying more money to elite athletes have been vital to improving the sporting performance of the race over the years. Indeed, Beatrice Foods raised, exponentially, the prize money from USD 77,000 in 1982 to USD 285,000 in 1986 (see Bank of America 2022 to examine the evolution of prize money paid to top athletes from 1982 to 2019) (see Table 2 for updated figures for the 2022 forthcoming race). Joan Benoit, one of the best female runners, valued the increase in the price money by stating "Beatrice has turned America's Marathon into the world's marathon. This is where the competition is" (Treadwell, 64, in Suozzo 2006, p. 23).

After Beatrice Food left the race, Old Style (Heileman Brewing Company) became the main sponsor for a short period of time (1988–1990), but the relationship finished after the company had financial problems and the company paid only half of the agreed-upon USD 1.2 million to sponsor the race (Suozzo 2006). After two years with no sponsor (1991–1993), from 1993 to the present day, banks (LaSalle from 1994 to 2007 and Bank of America from 2008 to present times) have become the main sponsors at the Chicago Marathon. These types of sponsors have been regarded as critical to the financial growth and to promote the stability of the race, including increasing prize money and bonuses to male and female runners, and, not least, to give, finally, as Paula Kobe 2020b suggests, the Chicago Marathon a name (from LaSalle Bank to the current name of The Bank of America Chicago Marathon). Aligned with Paula Kobe, in the case of the Chicago Marathon, the name of the sponsors goes before the name of the race; a situation that is different to the Valencia Marathon. The main reasons that could explain the attraction of the sponsors to the Chicago Marathon has been, according to Suozzo (2006), a combination of this event increasing the image awareness of those brands and due to the economic potential of marathon runners (more than 45,000), which is extremely attractive for banks.

**Table 2.** Similarities and differences between both Chicago and Valencia Marathons in relation to management and financial factors since their first editions to current times.

| | The Chicago Marathon (Now Known as the Bank of America Chicago Marathon) | The Valencia Marathon (Now Known as the Valencia Trinidad Alfonso EDP) |
|---|---|---|
| First edition | 25 September 1977 | 29 March 1981 |
| Current Timescale | First Sunday of October | First Sunday of December |
| Status | One of the six World Marathons Major by Abbott Organization; certified race by World Athletics and Gold Label by IAAF | World-renowned marathon; certified race by World Athletics and Platinum Level by IAAF |
| Management organization model | Bank of America owned and managed the Chicago Marathon since 2008 | Sociedad Deportiva Correcaminos (owner and rights holder of the Valencia Marathon) since 1981 |
| Number of runners | From 4200 runners in 1977 to 45,932 in 2019 and to 44,000 in the forthcoming 2020 Chicago Marathon | From 800 runners in 1981 to 30,000 in 2022 (8400 are local runners and the rest (more than 21,600) come from other parts and abroad |
| Entry fees for runners Revenue from entry fees runners | USD 230 for U.S. residents and USD 240 for non-U.S. residents Around USD 10.5 million | First registered runners from the first 10,000 (EUR 60), from 10,001 to 20,000 registered runners (EUR 90) and for the rest until 30,000 registered runners (EUR 120) Around EUR 2.7 million |
| Best sporting performance | 2:03:45 (Dennis Kimeto, KEN) (2013) (M); 2:14:04 (F. Brigid Koskei, KEN) (2019) (F) Current world record | 2:03:58 (Evans Chebet, KEN) (2019) (M); 2:17:16 (Peres Jepchirchir, KEN) (2019) (F) |
| Prize bonuses for elite athletes | USD 75,000 (Winner); USD 55,000 (Second); and USD 45,000 (Third) (Men and Women). Overall prize bonus: USD 740,500 (2022) Wheelchair division men/women: USD 25,000 (Winner); USD 18,000 (Second) and 3rd: USD 12,000 (Third) | USD 75,000 (Winner); USD 45,000 (Second) and USD 30,000 (Third) |
| Main corporate sponsor Sporting corporate brand Official sponsors Associate sponsors Institutional Partners | Bank of America since 2008 to present times Nike since 2008 Abbott, Advocate Health Care, Briofreeze, Endurance, Nike, Tata Consulting Services and Wanda Sport) Supporting (Goose Island, MacDonald, Athletics, IFIT) Blue Plate, Cultural Link, Chiquita, Deloitte, Hilton Chicago, Jewel Oso, Michigan Apples, Stryker, AED Superstore, Millennium Garages, PODS Media Partners (Chicago 5 NBC Chicago, Chicago Tribune, the 670 Score, Telemundo Chicago and 93 XRT) | Fundación Trinidad Alfonso since 2012 to present times New Balance since 2019 Caixa Bank, Coca-Cola, Zurich, Movistar, MSC Isaval, Vithas, Ecoembes, Yamaha, Prosegur Security, Bertolin Group, Marathon-photos.com, Farmacia Ribera Valencia City Council, Valencia Autonomous Community, World Athletics, AMIS, Spanish Athletic Federation, Renfe and Red Cross |
| Estimated spectators | 1.7 million | 200,000 |

Source: Updated information from Llopis-Goig and Paramio-Salcines (Llopis-Goig and Paramio-Salcines); Bank of America (2022); personal communication from the Bank of America Chicago Marathon communication department (17 July 2022) and Foundation Trinidad Alfonso and Valencia Ciudad del Running web page.

As with Old Style's (Heileman Brewing Company) short relationship with the Chicago Marathon, in the Valencia Marathon, Divina Pastora, an insurance company, became one of the main sponsors for a short period of time (from 2011 to 2012). However, the turning point in terms of sponsorship for the growth and expansion of the Valencia Marathon brand as it is known today (Marathon Valencia Trinidad Alfonso EDP), came in 2012 with the Fundación Trinidad Alfonso (Foundation Trinidad Alfonso), a non-profit foundation run by one of the main businessmen in Valencia and in Spain, Juan Roig. Foundation

Trinidad Alfonso became a sponsor of the Valencia Half Marathon at first, and since 2013, it has been the main sponsor of the Valencia Marathon and other races at a local level. The Foundation Trinidad Alfonso has been crucial in the evolution of the race in terms of the growth of runners, for their financial contribution to the Half Marathon and the Marathon along with the attraction of more sponsors to both races, the prize bonuses paid to runners based on their sporting performance in the last decade and to the expansion of running races in the city (Capsi and Llopis-Goig 2021). As different sources have estimated, the Valencia Marathon's organizers spent EUR 5.4 m for the budget of the 2019 edition and nearly EUR 1 m (around 20 percent of the budget), more than the prize paid to top athletes at the Chicago Marathon estimated at USD 841,500 in 2019 (Bank of America 2022), which has been reduced to USD 740,500 in 2022, to pay and attract the best male and female runners (personal communication, 15 July 2022). At the time of writing, and different to the Chicago Marathon, the Foundation Trinidad Alfonso represents the main sponsor by covering nearly half of the budget of both races, the Valencia Marathon, and the Valencia Half Marathon, while the rest, EUR 3.24 million (58 percent of the running cost of the race) comes from other sponsors (estimated at 24 companies, including New Balance, Coca-Cola, Zurich, Movistar, MsC, among others) (Fundación Trinidad Alfonso 2022) (Table 2). These sponsors also cover the Half Valencia Marathon with a budget of EUR 1.6 million.

Following key decisions (Collis 2021) and the steps suggested by Johnson et al. (2017), we have identified the following key decisions that help us to identify the similarities and differences between both marathons and, by extension, to identify what the competitive advantages are of these marathons. In terms of critical decisions, one similarity is the current location of both marathons in central and emblematic areas of both cities (Grand Park at the Chicago Marathon and City of Arts and Science Centre, designed by one of the mega start architects, Santiago Calatrava, at the Valencia Marathon) as already mentioned, which has clearly benefited both marathons. The change to central locations of the start and finish lines of both marathons can be considered an important organizational decision.

As Frick et al. (2019) argue on a large analysis of this interrelation, one of the incentives to increase the sporting performance of runners is the prize money paid for top runners in some of the world marathon majors (data from 2017) and the Valencia Marathon (data from 2021). Table 2 shows the main similarities and differences between the Chicago and Valencia Marathons in terms of their management factors and contributions from sponsors and their implications for the financial stability of both races since their inaugural editions. In terms of the prize money, the Chicago Marathon has gradually increased this in relation to management and financial factors since its first edition to current times. The Boston Marathon leads with the highest economic incentives, followed by the Chicago and New York Marathons, while the Valencia Marathon has been behind this prize money. However, the Valencia Marathon has quickly imitated the strategy of other leading marathons by increasing the prize bonuses paid to elite runners over the last decade; a decision that has contributed to improve substantially the sporting performance of runners, the MVTA's international reputation, to increase runners' participation and, not least, to drive the growth of local tourism.

The current MVTA DEP men's race fastest time (2:03:58) set by the Kenyan Evans Chebet in 2019 is the fourth-fastest in the marathon ranking after the BMW Berlin Marathon (2:01:39), the TCS London Marathon (2:02:37) and the Bank of America Chicago Marathon (2:03:45), which was set in 2013 (Bank of America 2022). Meanwhile, the Chicago Marathon has been credited with setting the women's world record (2:14:04) in 2019, followed by the TCS London Marathon (2:15:25) and, shortly after, the Valencia Marathon women's race fastest time (2:17:16) set by the Kenyan athlete Peres Jepchirchir. The sporting performance in all races run by SDC has contributed not only to the attraction of elite runners (with the Valencia "our personal best program" aiming to attract elite runners around the world, giving a prize bonus to those international runners that run sub 2 h 20′ (EUR 4000) or sub 2 h 45′ (EUR 2000), but also to increase the participation of local runners. This finding was in line with previous studies on marathons and outstanding performance (Frick et al. 2019).

The Chicago Marathon's runners' participation increased by nearly 450 percent in a short period (from 1995 to 2003), which contributed to the consolidation of the race as one of the leading marathons, and the Valencia Marathon has followed a similar pattern by increasing runners' participation by more than 400 percent in eleven years (from 2011 to 2022) from 6732 in 2011 to 25,546 in 2019, but with a substantial decrease in runners, due to the COVID-19 pandemic, to 12,668 in 2021. For the 2022 edition, the Valencia Marathon's participation will peak to a larger field size of 30,000 runners. Though this increase in runners in the Valencia Marathon in a short time is remarkable, one of the competitive advantages of the Chicago Marathon over its counterpart is that the Chicago Marathon attracted more runners (45,932) in the 2019 edition than the forthcoming 2022 Marathon Valencia Trinidad Alfonso. From a comparative perspective, the Valencia Marathon is still clearly behind the TCS London Marathon (59,632) (set up in the 2021 race), regarded as the world's biggest ever marathon, followed by the TCS New York Marathon (53,520), BMV Berlin Marathon (46,983), Boston Marathon (35,868) and Tokyo Marathon (35,460) (Llopis-Goig and Paramio-Salcines).

As in the case of the Chicago Marathon, participation in the Valencia Marathon by individuals from abroad has continued to grow in a short time, going from 769 participants in 2011 to 11,235 in 2019 and to 14,400 in 2022—and they came from a total of 115 countries. Similarly, the participation of Spanish runners from outside Valencia has also seen a tremendous growth, going from 2781 in 2011 to 12,508 in 2019 and to 7200 in 2022. Although around 8400 are local runners, more than 21,600 runners are coming from other parts and abroad. In essence, the MVTA EDP itself has become a significant sports tourism product. Regarding this relationship between performance in sports events and the intention to revisit the hosted city, this finding was in line with previous studies by Gholipour et al. (2020), Huang et al. (2015) and Park et al. (2019). In our case, the outstanding performance at the Valencia Marathon can be seen as directly affecting the growth of local tourism. This is also seen in the longitudinal ex-post studies carried out in recent years by the IVIE, in which it is concluded that the income generated by the activities that revolve around the celebration of the Valencia Marathon are much greater than the expenditures of the host organization.

The latest study (IVIE 2020) estimated that the operating costs of the 2019 Valencia Marathon amounted to EUR 5.4 million, while the income generated by the money spent in Valencia by runners and their companions amounted to EUR 22.8 million. Considering the expenditure is susceptible to general economic impact, the report indicates that, for every euro spent in organizing the event, EUR 4.2 more is generated through tourist spending. This is a clear example of the multiplier effect and the ultimate positive financial impact the event has on the Valencian economy. According to the event typology developed by UK Sport, the MVTA EDP is a clear example of a "competitor-driven event" or "participant event" as most of expenditure is led by the spending of runners.

## 5. Conclusions

In this paper we have analyzed critical aspects considered decisive in the evolution and success of the Chicago and Valencia Marathons over time. The importance of contemporary marathons goes beyond the sporting aspect, in a context in which main cities compete to attract the world's attention and seek something more than direct or indirect short-term effects of an economic nature. In the case of these two cities, the evolution and growth of their marathons has run parallel to strategies to reposition the city's image in a context of industrial decline. On the other hand, this growing competition has led to make the planning, execution, and management of any increasingly complex, contemporary marathon, given that, as some authors have pointed out (Collis 2021), there is no sequence or process that allows any marathon to be managed and planned linearly. This justifies the interest in understanding the aspects that play a critical role in its evolution and success.

While most of the research on marathons has focused on the economic dimension of marathons, our work has –as we have already pointed out– paid attention to key aspects of

the evolution and success of both events, in terms of the main management decisions taken over time, the way in which organizational leadership of the event has been exercized (Cooper 1992; Schwarz et al. 2017) and the financial support it has received from sponsors. The analysis of these aspects has been carried out taking as point of departure the socio-economic and political context of each city during the period of the event (Fox et al. 2014) and adopting a holistic perspective in the analysis of the event. This has allowed us to explore the interrelationships between different areas to consider them as part of a complete landscape strategy (Collis 2021; García-Vallejo et al. 2020).

The analysis undertaken allows us to conclude that the international success of both events –in terms of sporting participation, performance and economic impact– is closely related to the design and management of the event; the synergies between the political, business and sporting spheres that the organizational leadership of the event has favored their implementation and, as a consequence of all this, the financial contribution from sponsors, which has provided them with financial stability necessary to improve the management of the race.

Regarding the design- and management-related aspects of the race, the study noted the importance of the date fixed for the race, given its implications in terms of weather conditions and its place in the national and international marathon calendar, and the location of the start and finish of the race in the city center, insofar as it contributes to reinforcing both the spectacular nature of the race and the participants' experience and level of satisfaction.

As Cooper (1992) highlighted the critical role played by Fred Lebow in the evolution of the New York City Marathon or Suozzo (2002,2006) the contribution of Carey Pinkowski, the executive race director, to the growth of the Chicago Marathon, our study also highlighted the importance of the organizational leadership of the event. This leadership is especially necessary to manage the relations and synergies of the event with the municipal authorities, with the business sector and with the sporting world. The municipal authorities are essential to the extent that the organization of a marathon needs their support to promote the event. In addition, the municipal authorities play a central role in urban transformation and in the promotion of sporting culture. But, on the other hand, the marathon is a way for them to project themselves to the outside world, enhance the city's image and boost its tourist attractiveness, as the two cases analyzed show. The business sector is irreplaceable in its role as marathon sponsor. Firstly, because they provide the financial stability needed for this type of event. Secondly, because they increase the reputation of the event and help it to attract more elite runners, which is to the sporting performance's advantage. Thirdly, because their contribution goes beyond the economic injection and the financial reactivation they provide. They make us rethink the management culture which, for example, favors the establishment of a more competitive incentive system for runners.

The synergies with the world of sport are fundamental both in terms of attracting international talent and in relation to the local sporting culture. The results obtained reveal that both city marathons analyzed owe part of their growth and increasing prestige both to the records achieved by their winners and to the strengthening of the local running culture that has turned each edition of the event into a truly massive celebration.

There are at least two avenues for further research. First, the analysis of key decisions taken by senior managers and race directors, either alone or in partnerships with urban leaders and corporate sponsors, of other marathons over an extended period will be extremely useful to explain the sustained growth and evolution of city marathons as we analyzed in our study. Second, and considering that for instance the Singapore Marathon is under consideration by AbbottWMM to become the 7th World Marathon Major; race that can emulate or supersede those services offered at the Marathon Valencia Trinidad Alfonso EDP and the Bank of America Chicago Marathon, further studies might investigate the dynamic aspects of decision making of more cases of city marathons from different parts of the world. Finally, the present study is affected by some limitations that need to be pointed out. Firstly, the information we could access about each of the two marathons

was not always homogeneous and comparable. Obviously, the organizing bodies of both races disseminate and publicize different aspects of the events they manage with their own interests in mind, but not considering the sports management researchers' needs. Secondly, despite being two internationally renowned events, the marathons studied have very different dimensions, both in terms of the number of participants and economic impact, differences that also apply to the size and characteristics of the hosting cities.

**Author Contributions:** J.L.P.-S. and R.L.-G. contributed equally to this manuscript. All authors have read and agreed to the published version of the manuscript.

**Funding:** This research received no external funding.

**Data Availability Statement:** Data sharing not applicable.

**Acknowledgments:** The authors appreciate the comments from staff from the Bank of America Chicago Marathon. We will like to thank both reviewers for their insightful comments on the paper, as these comments led us to an improvement of the study.

**Conflicts of Interest:** The authors declare no conflict of interest.

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
