# Peer review of "Key Strategic Decisions and Their Influences on the Management and Success of the Bank of America Chicago Marathon and the Marathon Valencia Trinidad Alfonso"

_ijfs, doi:10.3390/ijfs10030074_

Round 1

Reviewer 1 Report

TITLE: Financial stability and corporate sponsors and their influences on the success of the Bank of America Chicago Marathon and the Marathon Valencia Trinidad Alfonso

I appreciate the chance to serve as a reviewer on this paper. The paper is well written and suits the scope of the issue. I urge the author(s) to consider the following comments and improve the paper accordingly.

1.      At first, I think the authors should make clearer the economic meaning of their research starting with an economic motivation.

2.      What are the economic effects by these results for the countries (especially in income of citizens).

3.      My main objection concerns the lack of empirical methodology. It would be useful if the authors use a methodology that measures the effects of their suggestions (see for instance Tsagkanos et al. 2021).

4.      It would also be useful for the audience and future researchers if a guide for the future research is provided: how this research could be used concretely to open new pathways? Is it possible to provide some examples and possible directions for future research?  

5.      I recommend checking English spelling and edit the paper carefully.

Literature

Tsagkanos, A., Gkillas, K., Konstantatos, C., Floros, C. (2021)Does volume drive systemic banks’ stock return volatility? Lessons from the Greek banking system.” International Journal of Financial Studies. Vol 24(9), 1-13

Author Response

Manuscript number: ijfs-1844045.

Dear editor and reviewer 1

First, we would like to thank the reviewers of our paper entitled ‘Financial stability and corporate sponsors and their influences on the success of the Bank of America Chicago Marathon and the Marathon Valencia Trinidad Alfonso’ for their constructive and positive assessment of our work and details suggestions. Please see our direct responses to your recommendations in the table below and is indicated in the text of the document by the coloured text.

Response to Reviewer 1 Comments

Point 1. At first, I think the authors should make clearer the economic meaning of their research starting with an economic motivation.

Response 1:

The reviewer´s point is well taken. This is an aspect that we have already addressed in the paper. We expect that this quote contributes to draw attention to readers of the journal as well as academics, and city politicians and managers about the relevance of this growing market.

In our research, we have found that only Running USA (https://www.runningusa.org/) offers a comprehensive value of city marathons in the US market. In our case, we used the information provided by this organization and stated in Leisure Business Market Research Handbook 2017-18 (6th edition) edited by Miller and Washington.

In a quest to understand the economic motivation of cities for hosting marathons, Running USA, a non-profit organization committed to the growth and success of the running industry, stated, “cities are embracing marathons for the economic upswing. One of the benefits of a marathon of any size is that it brings people to your city, it showcases your city, it brings people back” (in Miller and Washington 2017, 277).

At the same time, we have rewritten some parts of the introduction to condense the literature review sources and interest in this global phenomenon as city marathons represents. See those changes on yellow. See Page 2, lines 60 to 65

Point 2: What are the economic effects by these results for the countries (especially in income of citizens).

Response 2: As we said in the text, our work does not focus on the analysis of the economic effects of marathons on cities and countries that host them. This aspect is something that is made clear from the beginning and in the text. In the paper, we included the following statement: over the years, more financial data on some of the leading world city marathon has been made public.

In essence, our study aims to explore the reasons, that is, the decisions that have played a key role in the evolution and success of some of today's major marathons, specifically two of them, the Chicago and Valencia marathons. This task has been very scarcely treated by the literature specialized in this kind of events, so we believe it is a relevant contribution.

Unfortunately, there are not available studies estimating the economic effect of city marathons at national level. In the previous reply, we said that we have found that only studies undertaken by an organization such as Running USA (https://www.runningusa.org/) offer a comprehensive value of city marathons in the US market. See page 16. Lines 215-216

Point 3: My main objection concerns the lack of empirical methodology. It would be useful if the authors use a methodology that measures the effects of their suggestions (see for instance Tsagkanos et al. 2021).

Response 3: We understand the need to explain the methodology used. However, considering the aim of the paper we feel that our data collected will allow for replicability of the analysis.

Although we appreciate your suggestion, to measure the effect of those decisions taken empirically was more difficult. Instead, we relied on multiple sources as we explained below.

All the data that appears on our paper as we mention comes from a documentary analysis of an extensive number of academic and professional publications, annual reports from both Chicago and Valencia city councils and city marathons´ organizers, their official websites, research undertaken by economic groups in both cities (in the case of the Valencia Marathon, the main economic and tourist indicators come from the Instituto Valenciano de Investigaciones Económicas and we have also mentioned that we got further information from personal communications with staff from the AbbottWMM and the Bank of Chicago Marathon.

This multi-method data collection methodology produced a vast amount of rich data that has helped us to address the main objective of the paper. Also, to be consistent with the final objective of the paper, we have also modified the final paragraph.

The final text should read as:

This paper presents the main findings of both case studies based on content analysis of an extensive array of academic and professional publications, annual reports from both city councils, internal documents produced by both city marathons’ organizers from their inaugural editions to nowadays (the Chicago Marathon (then the Mayor Daley Marathon) started officially, for the second time on September 25, 1977, while the Valencia´s first edition was on March 29, 1981), annual reports undertaken by economic groups in those cities and personal communications with staff from the AbbottWMM and the Bank of America Chicago Marathon. Specifically, this paper analyzes the similarities and differences of the key strategic decisions taken by senior managers and race directors of both races over the time mentioned, focusing on inter-dependent factors influencing management such as the type of organizational structure, the best sporting performance, ratio of elite and amateur runners, their economic, sporting, social and tourist impacts, and other indicators (number, type, length and economic contributions from sponsors, estimated spectators, number of participants from all over the world, prize money to winners or budget of the management of the race).

Point 4: It would also be useful for the audience and future researchers if a guide for the future research is provided: how this research could be used concretely to open new pathways? Is it possible to provide some examples and possible directions for future research? 

Response 4. In our final paper submitted, we did not want to increase its length so substantially.

Trying to incorporate your comment we have considered to include a paragraph to cover this issue. See the final text on yellow

Considering the growth of the economic, social, sporting, urban and symbolic relevance of urban marathons worldwide and trying to address the limited studies that have analyzed a gap in the literature as the key decisions taken by senior managers and directors that explain the growth and evolutions of marathons, we have included a paragraph to inform about two further directions for research.

Page 16. Lines 695-703

There are at least two avenues for further research. First, the analysis of key decisions taken by senior managers and race directors, either alone or in partnerships with urban leaders and corporate sponsors, of other marathons over an extended period will be extremely useful to explain the sustained growth and evolution of city marathons as we analyzed in our study. Second, and considering that for instance the Singapore Marathon is under consideration by AbbottWMM to become the 7th World Marathon Major; race that can emulate or supersede those services offered at the Marathon Valencia Trinidad Alfonso EDP and the Bank of America Chicago Marathon, further studies might investigate the dynamic aspects of decision making of more cases of city marathons from different parts of the world. Finally, the present study is affected by some limitations that need to be pointed out. Firstly, the information we could access about each of the two marathons was not always homogeneous and comparable. Obviously, the organizing bodies of both races disseminate and publicize different aspects of the events they manage with their own interests in mind, but not considering the sports management researchers’ needs. Secondly, despite being two internationally renowned events, the marathons studied have very different dimensions, both in terms of the number of participants and economic impact, differences that also apply to the size and characteristics of the hosting cities.

Point 5. I recommend checking English spelling and edit the paper carefully.

The point is well-taken. We have reviewed the paper to correct some mistakes that we have found. This aspect has already addressed in the new version of the paper. See those changes on yellow

We hope that you find the proposed recommendations for the submitting of our paper satisfying.

Your sincerely

Juan L. Paramio-Salcines and Ramón Llopis-Goig.

Reviewer 2 Report

I found some value in this manuscript. It deals with a  interesting research issue analyzing the key strategic decisions affecting the growth and evolution of two city marathons. Research in this area is in relative short supply.

However I suggest that authors better clarify the objective of their work as they mainly deal with the strategic decisions affecting the events. I suggest to declare that the main objective of the paper is to analyze these strategic decisions while financial stability provided by sponsors is only one aspect of their analysis. Accordingly I suggest to rework the title and the abstract of the paper through emphasizing the true objective of the analysis that is dealt with. 

I also suggest to highlight the study limitations which I did not find in the final sections.

Author Response

Manuscript number: ijfs-1844045.

Dear editor and reviewer 2

First, we would like to thank the reviewers of our paper entitled ‘Financial stability and corporate sponsors and their influences on the success of the Bank of America Chicago Marathon and the Marathon Valencia Trinidad Alfonso’ for their constructive and positive assessment of our work and details suggestions. Please see our direct responses to your recommendations in the table below and is indicated in the text of the document by the colored text.

Point 1. I suggest that authors better clarify the objective of their work as they mainly deal with the strategic decisions affecting the events. I suggest declaring that the main objective of the paper is to analyse these strategic decisions while financial stability provided by sponsors is only one aspect of their analysis. Accordingly, I suggest to rework the title and the abstract of the paper through emphasizing the true objective of the analysis that is dealt with.

Response 1. Thank for your comment. We have modified the title of the paper, keywords and the abstract focusing on the strategic decisions taken by senior managers to reinforce the relevance of the study.

The final title will appear as

Key strategic decisions and their influences on the management and success of the Bank of America Chicago Marathon and the Marathon Valencia Trinidad Alfonso

In the new version of the abstract the text should read as the following: see the major changes in yellow

Abstract: City marathons have evolved and grown exponentially in type and popularity, managerial complexity, for their financial impact on their host cities and for the attraction of corporate sponsors. Most of the research on city marathons has focused on evaluating their broad economic, urban, tourist, social, sporting, and symbolic effects on host cities. However, less attention has been paid to analyzing key strategic decisions that could account for the evolution and growth of specific marathons and their influences on their management and success. This article, which addresses the cases of the Bank of America Chicago Marathon and the Marathon Valencia Trinidad Alfonso, examines those key strategic decisions that have been taken from their inaugural first editions to present and how effective they have been as regards the management and success of both races. Results show that the international success of both events –in terms of sporting participation, performance, and economic impact– is closely related to critical key decisions taken to improve the design and management of the event; the synergies between the political, business and sporting spheres that the organizational leadership of both races has favored their implementation and, as a consequence, the support received from sponsors. This factor has not only provided both races with financial stability, but it has also contributed to improving how both marathons are managed.

Point 2. I also suggest highlighting the study limitations which I did not find in the final sections.

Response 2. In principle, we did not want to increase the length of the article so substantially, but we have included a paragraph to inform about study limitations and future areas of research.

See page 16. Lines 706-714

There are at least two avenues for further research. First, the analysis of key decisions taken by senior managers and race directors, either alone or in partnerships with urban leaders and corporate sponsors, of other marathons over an extended period will be extremely useful to explain the sustained growth and evolution of city marathons as we analyzed in our study. Second, and considering that for instance the Singapore Marathon is under consideration by AbbottWMM to become the 7th World Marathon Major; race that can emulate or supersede those services offered at the Marathon Valencia Trinidad Alfonso EDP and the Bank of America Chicago Marathon, further studies might investigate the dynamic aspects of decision making of more cases of city marathons from different parts of the world. Finally, the present study is affected by some limitations that need to be pointed out. Firstly, the information we could access about each of the two marathons was not always homogeneous and comparable. Ob-viously, the organizing bodies of both races disseminate and publicize different aspects of the events they manage with their own interests in mind, but not considering the sports management researchers’ needs. Secondly, despite being two internationally renowned events, the marathons studied have very different dimensions, both in terms of the number of participants and economic impact, differences that also apply to the size and characteristics of the hosting cities.

We hope that you find the proposed recommendations for the submitting of our paper satisfying.

Your sincerely

Juan L. Paramio-Salcines and Ramón Llopis-Goig.

Round 2

Reviewer 2 Report

NONE